# Molecular Characterization of a New Ecotype of Holoparasitic Plant *Orobanche* L. on Host Weed *Xanthium spinosum* L.

**DOI:** 10.3390/plants11111406

**Published:** 2022-05-25

**Authors:** Ali Reza Yousefi, Asadollah Ahmadikhah, Reza Fotovat, Leila Rohani, Foad Soheily, Daniela Letizia Uberti, Andrea Mastinu

**Affiliations:** 1Department of Plant Production & Genetics, University of Zanjan, Zanjan 45371, Iran; r_fotovat@znu.ac.ir (R.F.); leila_rhn@yahoo.com (L.R.); soheilyfoad@gmail.com (F.S.); 2Department of Plant Sciences and Biotechnology, Faculty of Life Sciences and Biotechnology, Shahid Beheshti University, Tehran 31587, Iran; a_ahmadikhah@sbu.ac.ir; 3Department of Molecular and Translational Medicine, University of Brescia, 25123 Brescia, Italy; daniela.uberti@unibs.it

**Keywords:** *Orobanche aegyptiaca*, phylogeny, xanthium, 5.8S rRNA

## Abstract

A species of *Orobanche* was observed on spiny cocklebur (*Xanthium spinosum*) for the first time in Iran and tentatively was named IR-Iso.This study was conducted to make a phylogenetic analysis of the *Orobanche* using 5.8S rRNA region sequences, and also to better understand its sequence pattern. The full-length ITS1-ITS2 region of the new *Orobanche* isolate was PCR-amplified from the holoparasitic plant parasitizing *X. spinosum*. Sequences of the amplicons from the isolate were 100% identical but differed by 5.6–6.7% from most homologous GenBank accessions to 37.9% divergence from distant species. The analysis of the molecular variance showed that variation between-population (61.9%, SE = 0.04) was larger than within-population. Neighbor-joining analysis placed the Iranian isolate in the same clade as most of the *Orobanche* and *Phelipanche* species. The isolate was more closely related to *Orobanche aegyptiaca* (from China), and this was confirmed by using a structure analysis. However, complementary analyses showed that the Iranian isolate has a unique nucleotide substitution pattern, and hence it was considered as an ecotype of *O. aegyptiaca* (ecotype *Alborzica)*. In this paper we report on the association between this new ecotype of *Orobanche* and *X. spinosum*.

## 1. Introduction

The Orobanchaceae family is the largest family of parasitic flowering plants, including nearly 2060 holo- or hemiparasitic species in 90–99 different genera [1,2,3,4]. Among them are many taxa that are not included in the Crop Wild Relatives lists in many countries, and due to their restricted distribution, they should be given a higher priority for conservation [5]. However, this plant family also includes species that occur in arable lands and that could threaten crop production. *Orobanche* is the most agriculturally important genera of Orobanchaceae and contains around 200 species distributed all over the world [3]. In Iran, there are 36 different *Orobanche* speciesthat have been recorded [6]. All plants in the genus *Orobanche* are root parasites that naturally attach to their host with haustoria formation to extract nutrients [7,8]. *Orobanche* species have been identified in 58 countries [9]. Despite the widespread distribution of *Orobanche* members worldwide, yield loss of host crops occurs mainly in the north temperate regions, including East and South Europe, West Asia, and North Africa [7,9,10]. In the Mediterranean area and West Asia, *Orobanche* members threaten about 16 million ha of arable land, and crop yield losses due to *Orobanche* spp. attacks range from 5 to 100% [11,12,13,14]. 

In Iran, as in most other regions, *Orobanche*
*aegyptiaca* (Pers.) is one of the more widespread species and often attacks summer crops, especially Solanaceae (e.g., eggplant, potato, tobacco and tomato) [6]. Beside the *O. aegyptiaca*, other important species that are considered for research include: *O. ramose* (L.) Pomel (which often attacks to annual crops, such as tobacco, tomato, hemp, and cabbage), *O. crenata* Forssk (which grows on legume crops), *O. Cumana* Wallr. (which attacks Asteraceae crop plants, such as sunflower) [15], *O. gracilis* Beck. (which growson Fabaceae plants), *O. pancicii* Beck. (parasitizing *Knautia* L. and possibly also *Scabiosa* L. species), *Phelipanche lavandulacea* Pomel. (whose single perennial host is *Bituminaria bituminosa (L.) C. H. Stirt.*), and finally *P. purpurea* (Jacq.) Soja′k. (which often grows on Asteraceae plants such as *Artemisia* L. and *Achillea wilhelmsii* C. Koch) [15,16].

Our understanding of the phylogenetic relationships among major clades of *Orobanche* [17,18,19], along with species-level relationships, have been greatly advanced by studying molecular data [20,21,22]. 

Identification of *Orobanche* species is difficult because herbarium material lose their color and vegetative organs are reduced. Furthermore, the majority of *Orobanche* species are rare and endangered, and they do not have a clear taxonomic description as a result of their rarity [3].

A species of *Orobanche* was recorded on spiny cocklebur (*Xanthium spinosum* L.) for the first time. This study was conducted to make a phylogenetic analysis of this new isolate of the holoparasitic plant belonging to *Orobanche* genus using the sequences of ITS1-ITS2 region, and to better understand its sequence pattern in the studied rRNA region.

## 2. Results

### 2.1. Morphological Characters

The attachment of the collected sample of broomrape (Iranian isolate, IR-Iso) to spiny cocklebur (*X. spinosum*) roots was verified visually (Figure 1). The stems were erect (14–19 cm height), branched, glandular-pubescent and pale yellowish. The bracts were 0.5 to 0.6 cm long.

The flowers were surrounded by one bract and two bracteoles. The bracts were 0.4 to 0.5 cm long, and bracteoles measured 0.5 cm. The calyces were gamosepalous, 0.4 cm long, and glandular. Corolla were medium slate blue with darker veins, 1.8 to 2.0 cm, conspicuously infundibuliform, slightly curved, and glandular-pubescent. Stamens were epipetalous, inserted 0.5 cm above the corolla base, with hairy filaments, 1.1 to 1.4 cm long, and the anthers were villous. The style (measured 1.4 to 1.6 cm) and stigma (measured 0.5 cm) lobes were light steel blue. 

### 2.2. Sequence Analysis and Identities

5.8S rRNA fragments from the Iranian isolate were amplified using specific primers that bind to conserved regions of the ITS sequences. A unique PCR product of approximately 700 bp was generated (in two samples) and nucleotide sequences of the PCR products were determined by Sanger sequencing (613 nucleotides). Sequences from the two samples were 100% identical. The sequence (hereafter named IR-Iso) spanning the ITS1, 5.8S rRNA and ITS2 regions were aligned to other known Orobanchaceae sequences stored in GenBank. Seventy-one accessions in NCBI showed a significant similarity to IR-Iso. These 71 accessions, plus 11 additional accessions from the work of Frajmanet al. [23], formed our data set. Haplotyping analysis showed that 30 haplotypes (including the IR-Iso) existed in the data set (Table 1; Appendix A). TheIR-Iso differed from most similar GenBank accessions by 5.6–6.7%, but was more distant from the 11 additional accessions in [23] (from number 22 to 30 in Table 1).

The nucleotide frequencies from the sequences of the Orobanchaceae ITS1-ITS2 region in our data set (including out group sequences) are illustrated in Figure 2. It seems that significant differences exist in the frequency of A/T (21.9% vs. 24.2%) and G/C (26.4% vs. 27.4%) nucleotides. Also, G+C content (53.9%) is significantly higher than A+T content (46.1%). 

Distance analysis in MEGA6 (Appendix A) showed that in general IR-Iso showed minimum (5.9%) and maximum (41.3%) sequence divergence to KC811184 (*O. aegyptiaca* isolate 26-9S163TJGFQ2) and AY209282 (*O. anatolica* Boiss. &Reut. isolate 1) respectively, with overall divergence estimated at 11.96% (SE = 0.012).

### 2.3. Genetic Diversity Analysis

Maximum likelihood estimation (MLE) showed that the Kimura 2-parameter model (1980) was the best-fitted model for the nucleotide substitution pattern (Appendix A). Thus, genetic diversity was performed using the Kimura 2-parameter model.

MCL estimate of the nucleotide substitution pattern showed that transitional substitutions (bold and italicized values in Table 2) were significantly higher than transversional substitutions. The transition/transversion ratios were k_1_ = 3.69 (purines) and k_2_ = 5.95 (pyrimidines). The overall transition/transversion (R) bias was equal to 2.44.

### 2.4. Cluster Analysis

Thirty accessions, including IR-Iso, were classified by the neighbor-joining (NJ) method (Figure 3). Based on the NJ tree, two distinct groups were recognized with bootstrap values higher than 50%. High bootstrap support was obtained for the first group (99%), which contained two clades. Clade I has twenty-one Orobanchaceae species. As seen, most Orobanchaceae species were clustered together, and hence are monophyletic. An exception to this is the *Orobanche purpurea* (Jacq.), which alone formed a distinct clade in the basal side of the first clade in group I. As seen in Figure 3, the Iranian ecotype IR-Iso is co-clustered with *O. aegyptiaca* isolates 26-9S163TJGFQ2 from China. The seven accessions placed into group II have six clades.

AMOVA showed that the mean diversity in the entire population was 0.123 (SE = 0.013), which consisted of the inter-population diversity of 0.076 (SE = 0.012) and the diversity within subpopulations of 0.047 (SE = 0.004), forming 61.8% and 38.2% of total diversity, respectively. AMOVA also showed that between-population variation composed a large part of the genetic differentiation (61.9%, SE = 0.04).

The above grouping was confirmed using Bayesian structure analysis (Figure 4). It can be seen that there are two real groups in the studied population, which is revealed by using the Δ*K* method proposed by Evanno et al. [24] (top graph in Figure 4). The same accessions that form the two groups depicted in Figure 3 were assigned to distinct groups by the Bayesian method of structure analysis (bottom graph in Figure 4).

Tajima’s neutrality test showed that 37.9% of sites were segregated in the data set (Table 3), but only 9.6% of them were phylogenetically informative. Accessions in the data set showed a high similarity, as the overall mean genetic distance (π) in the data set was nearly 8.8%. Tajima’s D was equal to −0.304, indicating no deviance of mutation-genetic drift equilibrium; in other words, there is no evidence that the selection was a powerful force in the evolution of the studied accessions. In contrast, when the neutrality test was completed for group I, which consisted of 22 accessions, it was seen that only 21.8% of sites were segregated,6.9% of which was the nucleotide diversity, indicating the existence of more genetic similarity in this group. However, 25.2% of sites were segregated in group II, only 2.6% of which was the nucleotide diversity in the group.

## 3. Discussion

The ITS region of the 18S-5.8S-26S nuclear ribosomal cistron is now extensively used around the world for taxonomic classification, having been first utilized more than two decades ago [25,26]. Nuclear rDNA has hundreds to thousands of repeats in plant genomes, therefore, they are more easily isolated than most low-copy nuclear loci. Additionally, little experimental expertise is required for their successful amplification. The isolated sequence of ITS1-ITS2 region in this study contained 613 nucleotides. In general, the length of ITS sequences in plants has a relatively wide range (approximately 500–700 bp in angiosperms [27] to 1500–3700 bp in some gymnosperms [28,29]). It has been reported that this level of ITS sequence variation is suitable for phylogenetic inference at the specific, generic or even family levels [25,27]. Baldwin and other researchers concluded that the variation at hierarchical levels is related mostly to nucleotide polymorphisms, among which indel polymorphisms are common. 

In this study, we described the molecular characterization and phylogenetic analysis of the 5.8S rRNA genic region of a new isolate of *Orobanche*, associated with host weed *X. spinosum* in one locality of Karaj, Alborz Province, Iran. The ITS sequences were also used to confirm the identity of broomrapes in species *Aconitum lycoctonum* L. in the Alps [4]. 

It was supposed that because of the removal of ITS sequences during rRNA transcript processing (e.g., cleavage of the primary transcript within ITS-1 and ITS-2 during maturation of the small subunit, 5.8S, and the large subunit, ribosomal RNAs), they would be subject to mild functional limitations, which in turn would offer a prevalence of nucleotide sites evolving neutrally [30,31].

DNA sequence divergence data, particularly divergence in 5.8S rRNA, are widely used for defining species [19]. The IR-Iso isolate studied here showed 5.8–6.6% divergence from most homologous *Orobanche* species, and up to 37.9% divergence from distant species (Appendix A). In our work, the average divergence between the studied species was estimated at 12.0%, with 9.6% of them being phylogenetically informative. Therefore, a cut-off of 12% divergence is recommended as a criterion for demarcating species in this research. Baldwin et al. [27] reported divergence values ranging between 0–39% in pairwise comparisons between taxa, of which 5–59% was potentially phylogenetically informative. DNA sequence comparisons of the ITS1-ITS2 region sequence revealed ~82.7–94.5% identity with known *Orobanche* spp. and indicated that the Iranian isolate (IR-Iso) was more closely related to *O.*
*aegyptiaca*, and this was confirmed by using Bayesian structure analysis (Figure 3), although the disparity index test (Appendix A) showed that IR-Iso has evolved with a different pattern of nucleotide substitution relative to all other known *Orobanche* species.

*Orobanche aegyptiaca* is the most agriculturally important species of broomrape in Iran, and it is known to attack important crops and some ornamental plants, such as *Kalanchoe*
*blossfeldiana* (Poelln.) [32]. However, to our knowledge, this study reports the first occurrence of a spiny cocklebur plant as a host for *O. aegyptiaca*. Spiny cockleburis an annual weed that is widely dispersed around the world. Our findings suggest that spiny cocklebur plants could serve as a ″green ridge″, allowing *O. aegyptiaca* to grow and produce large quantities of seeds freely in the absence of other host crops (e.g., in fallow conditions or outside of fields), however, this in turn could increase the infestation level that occurs in the following years. Therefore, control of this wild host plant should be given a greater priority as a part of an integrated weed management system to avoid increasing the weed seed bank in the soil.

## 4. Materials and Methods

### 4.1. Plant Material and Sampling

Plant material of Orobanchaceae from Chaharbagh (35°50′20″ N 50°50′53″ E), a locality in Karaj, Alborz Province, Iran (Figure 5), was collected on a summer annual weed plant (*X. spinosum*). 

### 4.2. Extraction of DNA, PCR Condition and DNASequencing

DNA from leaf samples was extracted using the cetyltrimethyl ammonium bromide (CTAB) method described in [33]. Amplification and sequencing of the given DNA region was completed using the method described by Schneeweiss et al. [19]. DNA sequencing was performed by using an *ABI* automated sequencer (Bioneer Co., Seoul, Korea).

### 4.3. Sequence Analyses and Phylogenetic Studies

BLASTn searches of the GenBank ‘nr’ database was used for initial sequence identification. Sequence alignment was achieved using multiple sequence alignment software ClustalW. The data set consisted of 19 sequences from NCBI that showed a significant similarity with our query of sequences 613 nucleotides in length, obtained by using the BLAST tool, and an 11 additional accessions studied by Frajman et al. [23]. The accessions with GenBank numbers in the molecular analysis are given in Table 1. For building the phylogenetic tree, firstly the model selection was performed in MEGA6 software [34]. Subsequently, the model with the lowest Bayesian Information Criterion (BIC) score was considered the best model for the description of the substitution pattern. The evolutionary history was inferred using the neighbor-joining method [35] in MEGA6. A non-parametric bootstrap analysis with 5000 replicates provided quantitative support for recovered nodes [36]. Branches that hadless than 50% bootstrap replicates were collapsed. Distance and pattern analysis tools of MEGA 6.0 software were used for further analysis of the sequences. *O. anatolica isolate 1* was used as an outgroup, because it had the longest branch in the preliminary phylogenetic tree.

For the estimation of the substitution matrix, substitution patterns and rates were estimated using the TamuraNei [37] model. The relative instantaneous *r* values were calculated. An estimation of the pattern of nucleotide substitution and rates was performed usingthe Kimura 2-parameter model [38]. The molecular clock was tested by comparing the ML value for the given topology both with and without the molecular clock constraints usingthe above-mentioned model [38].

## 5. Conclusions

Based on the molecular features of the given 5.8S rRNA region, such as nucleotide similarity, disparity index test and phylogenetic analysis, it can be concluded that the Iranian isolate is a new separate isolate, as it has clear differences from other analyzed accessions as a whole, and even from the closest isolates of *O. aegyptiaca* that co-clustered with them. Therefore, it can be considered to bea new ecotype of *O. aegyptiaca,* because it has the capability of parasitizing a new host, *X. spinosum*. Considering the host specialty and molecular analyses, we propose IR-Iso is a new ecotype of *O. aegyptiaca* and suggest its scientific name as *O. aegyptiaca* ecotype Alborzica. To our knowledge, this is the first report of infestations of *O. aegyptiaca* on *X. spinosum*. This relationship can be important from two perspectives. Firstly since Orobanche is an obligate parasite plant, it can only grow in the presence of the host plants, therefore, in arable lands, the presence of *X. spinosum*, especially during the fallow year (land that is left unseeded during a growing season) can guarantee growth and seed production of *O. aegyptiaca*. Therefore, the parasite weed could pose a serious threat to crops, its occurrences during the fallow should be monitored and controlled using appropriate weed management practice to minimize the production of new seeds. Second; the presence of *X. spinosum* as a host plant in wild lands will allow *O. aegyptiaca* to grow freely, which will prevent the extinction of this species, and help preserve biodiversity.

## Figures and Tables

**Figure 1 plants-11-01406-f001:**
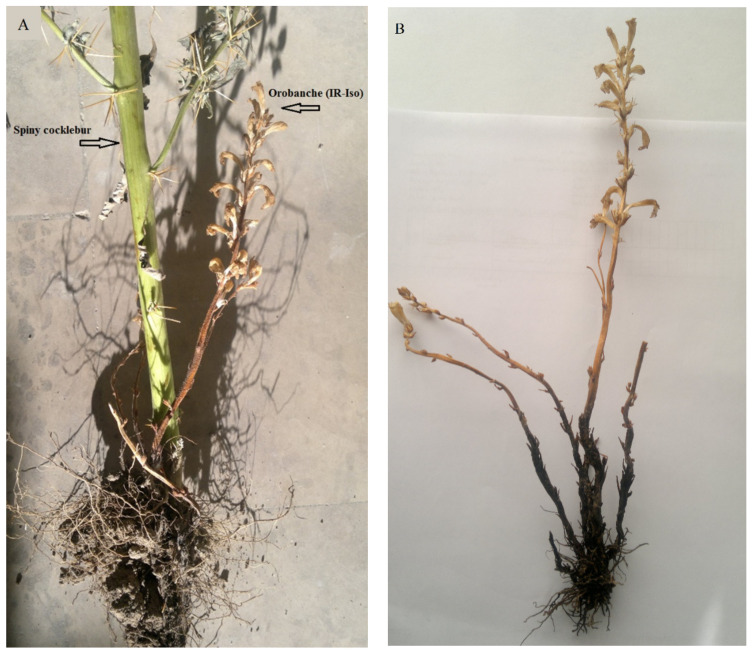
(**A**) The infection of a spiny cocklebur *(Xanthium spinosum* L.) by the collected broomrape (*Orobanche sp.*) from Iran. (**B**) *Orobanche* isolate detached from the roots of the host plant.

**Figure 2 plants-11-01406-f002:**
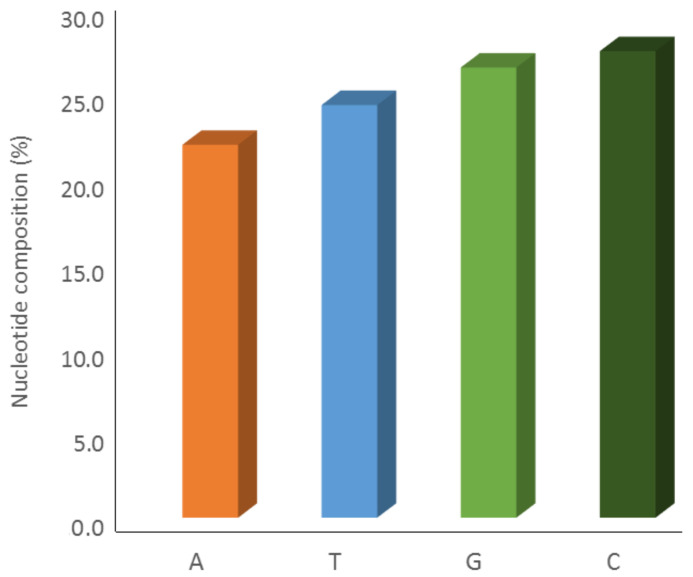
Nucleotide compositions of the 30 haplotype sequences in data set.

**Figure 3 plants-11-01406-f003:**
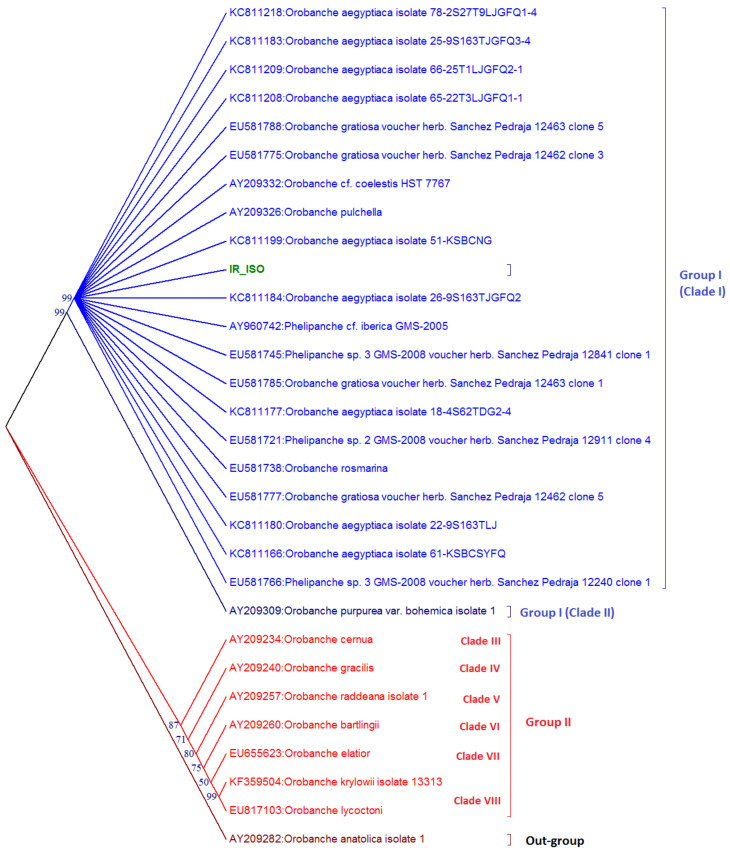
The unrooted tree of nuclear 5.8S rRNA sequences of Orobanchaceae reconstructed by the neighbor-joining (NJ) method considering *O. anatolica* as outgroup. Numbers at the nodes represent the bootstrap values over 50%. Iranian isolate (IR-Iso) is shown in green.

**Figure 4 plants-11-01406-f004:**
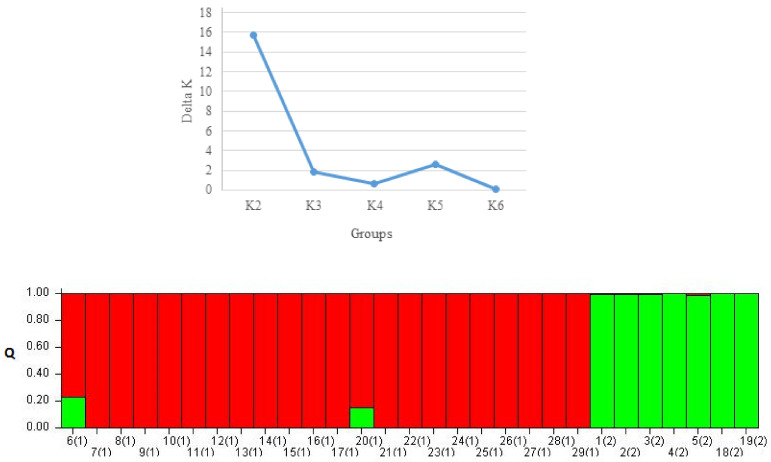
Bayesian-based clustering of the same individuals assigned to two groups by Mega 6. Real number of groups determined by Evanno et al. (2005) method (**top**). Graphical representation of the assignment of each individual to one of two red or green groups (**bottom**). Numbers in parenthesis on horizontal axis are group numbers determined by Mega 6.

**Figure 5 plants-11-01406-f005:**
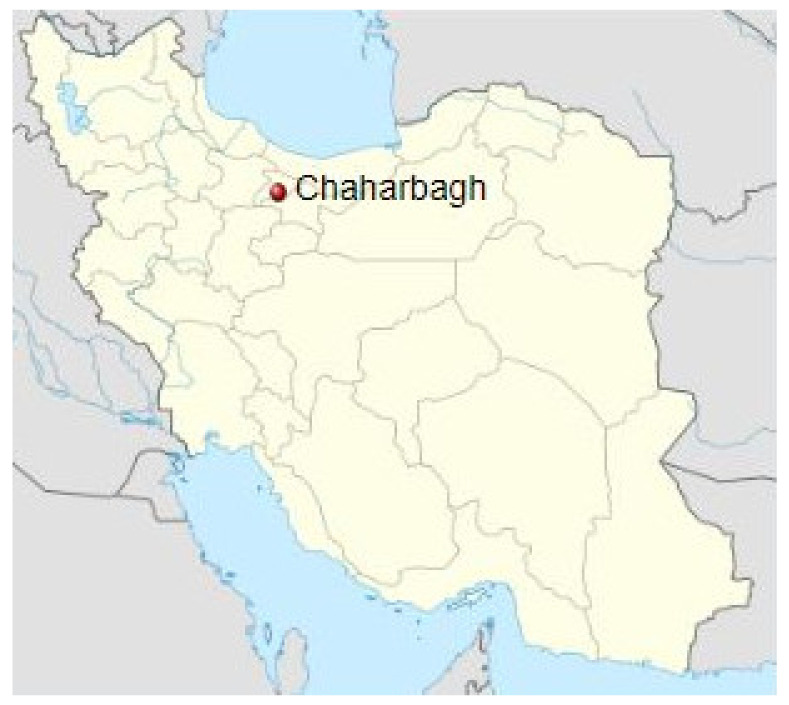
Map showing the region where material was collected in Karaj, Alborz Province, Iran.

**Table 1 plants-11-01406-t001:** Sequences showing similarity to IR-Iso ecotype after haplotype analysis.

ID	Accession Number	Scientific Name	Origin	Host Plant	IR-ISO
1	IR-Iso	*Orobanche* sp.	Iran	Spiny cocklebur	100.0
2	AY209326	*Orobanche pulchella*	Georgia	-	94.46
3	AY960742	*Phelipanche* cf. *iberica*	Turkey		94.30
4	EU581788	*Orobanche gratiosa*	Spain	-	94.30
5	EU581777	*Orobanche gratiosa*	Spain	-	94.30
6	KC811184	*Orobanche aegyptiaca*	China	Tomato	94.41
7	EU581785	*Orobanche gratiosa*	Spain	-	94.14
8	EU581775	*Orobanche gratiosa*	Spain	-	94.14
9	AY209332	*Orobanche* cf. *coelestis*	Turkey	-	94.14
10	KC811199	*Orobanche aegyptiaca*	China	Pumpkin	94.24
11	KC811218	*Orobanche aegyptiaca*	China	Tomato	94.08
12	KC811209	*Orobanche aegyptiaca*	China	Tomato	94.08
13	KC811208	*Orobanche aegyptiaca*	China	Tomato	94.08
14	KC811183	*Orobanche aegyptiaca*	China	Tomato	94.08
15	KC811180	*Orobanche aegyptiaca*	China	Chili	94.08
16	EU581766	*Phelipanche* sp.	Spain	-	93.81
17	KC811177	*Orobanche aegyptiaca*	China	Watermelon	93.91
18	EU581745	*Phelipanche* sp.	Spain	-	93.65
19	KC811166	*Orobanche aegyptiaca*	China	Tomato	93.77
20	EU581721	*Phelipanche* sp.	Spain	-	93.34
21	KF359504	*Orobanche krylowii*	Russia	-	77.37
22	EU817103	*Orobanche lycoctoni*	Slovenia	-	77.31
23	AY209260	*Orobanche bartlingii*	Croatia	-	77.07
24	AY209240	*Orobanche gracilis*	Morocco	-	76.42
25	AY209282	*Orobanche anatolica*	Turkey	-	93.65
26	EU581738	*Orobanche rosmarina*	France	-	76.97
27	AY209257	*Orobanche raddeana*	Georgia	-	76.26
28	AY209234	*Orobanche cernua*	Jordan	-	88.47
29	AY209309	*Orobanche purpurea*	Germany	-	76.19
30	EU655623	*Orobanche elatior*	France	-	77.13

**Table 2 plants-11-01406-t002:** Maximum composite likelihood estimate of the pattern of nucleotide substitution.

	A	T	C	G
A	-	*3.51*	*4.05*	**14.26**
T	*3.16*	-	**24.09**	*3.87*
C	*3.16*	**20.87**	-	*3.87*
G	**11.63**	*3.51*	*4.05*	-

Note: Each entry shows the probability of substitution (r) from one base (row) to another base (column) [17]. For simplicity, the sum of *r* values is made equal to 100. Rates of different transitional substitutions are shown in bold and those of transversional substitutions are shown in italics.

**Table 3 plants-11-01406-t003:** Tajima’s test statistics for the 5.8S rRNAplus ITS sequences of *Orobanche*.

	m	n	S	ps	θ	π	D
Group I	8	579	126	0.218	0.084	0.069	−0.945
Group II	22	584	147	0.252	0.069	0.026	−2.556
Overall	30	560	212	0.3791	0.096	0.088	−0.304

m = number of sequences; n = total number of sites; S = Number of segregating sites; ps = S/n; θ (functional coefficient of divergence) = ps/α1; π = nucleotide diversity; D = the Tajima’s test statistic (Tajima, 1989).

## Data Availability

The data presented in this study are available on request from the corresponding author.

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
