# Peer review of "Molecular Characterization of a New Ecotype of Holoparasitic Plant Orobanche L. on Host Weed Xanthium spinosum L."

_plants, 2022, doi:10.3390/plants11111406_

Round 1

Reviewer 1 Report

The text is misspelled in many places and is difficult to understand. Both tables and diagrams have errors at some points and are not always easy to understand. I believe that the text has serious problems. Extensive control by a native speaker should be presented and both the methodology and the conclusions should be presented in a more comprehensible way.

Author Response

We thank the Reviewer for his efforts in revising the manuscript. Now the text has been revised, we remain at your disposal for further clarifications and specific improvements indicated by the Reviewer.

Reviewer 2 Report

The authors propose a manuscript titled “Molecular characterization of a new ecotype of holoparasitic plant Orobanche on host weed Xanthium spinosum”. The scientific researchers in the last years have taken particular attention was given on topic aspect on Molecular characterization of new ecotypes of some parasitic plants as species of genus Orobanche, and in particular in the case proposed by authors on weed Xanthium italicum. The article is well structured and written but deserve of some correction that I have highlighted. I appreciate the original idea of the work which with a few revisions will convince me and the editor to publish it on Journal.

Title

Please consider in the title the scientific name in the complete way with the author name, in this case L. = Linneus. …..Orobanche L. on host weed Xanthium spinosum L.

Introduction

  • Lines 41-42. Please complete the concept with this crucial point and choose 3/4 international references: “The Orobanchaceae family as the largest family of parasitic flowering plants includes nearly 2060 holo- or hemiparasitic species in different genera [1], and are very important also for their conservation because are very rare and are not included in the Crop Wild Relatives lists in many countries [choose 3/4 references, one of these Perrino and Wagensommer 2022].
  • Line 42. See my previous comment. In general in the international audience the author of the species is reported only the first time in the paper. So, remember this my annotation for whole manuscript. Orobanche
  • Lines 44-45. When the author declare “In Iran there are 36 different Orobanche species [3]”, want to refer only to wild species or togheter cultivated and wild? Please clarify because in some countries Orobanche are also cultivated!
  • Lines 45-48. Please add references for these statements “All plants in the genus Orobanche are root parasites that naturally attach to their host with haustoria formation to extract nutrients (choose reference). Orobanche species were reported from 58 countries (choose 3/4 references). Despite widespread distribution of Orobanche members worldwide, yield loss of host crops occurs mainly in the north temperate regions including East and South Europe, West Asia and North Africa (choose reference);
  • Line 52. See my previous comments:
  • Orobanche aegyptiaca
  • Line 59. lavandulacea in italic
  • Line 59. See my previous comment. Bituminaria bituminosa (L.) C.H. Stirt.

  1. Results

Well done, the figure and tables are clear. Few observations

  • Table 1. Please must be separate genus to species in this way: i.e. Orobanche gratiosa, Orobanche aegyptiaca;
  • Line 160. Amova or Anova? Please verify

  1. Discussion

Few observations

  • Line 205. Space between gymnosperms[22, 23].
  • Line 235. See my previous comments for botanical point of view. Kalanchoe Adanson;

  1. Materials and Methods
  • Please give a geographic coordinates about material collection, of Karaj (Alborz Province, Iran) and if possible a map.
  • Line 245. All in italic: …Plant material and sampling;
  • Lines 246-247. Please check the font and size of characthers of this period: “Plant material of Orobanchaceae from one locality in Karaj, Alborz Province, Iran was collected on a summer annual weed plant Xanthium spinosum”.
  • Line 269. Orobanche anatolica

  1. Conclusions

Please spend two words about future research perspectives on these topic.

  • Line 279. aegeptiaca in italic

Reference to be added:

  • Perrino, E.V.; Wagensommer, R.P. Crop Wild Relatives (CWRs) Threatened and Endemic to Italy: Urgent Actions for Protection and Use. Biology 2022, 11, 193. DOI: 10.3390/biology11020193.

Author Response

REVIEWER#2

The authors propose a manuscript titled “Molecular characterization of a new ecotype of holoparasitic plant Orobanche on host weed Xanthium spinosum”. The scientific researchers in the last years have taken particular attention was given on topic aspect on Molecular characterization of new ecotypes of some parasitic plants as species of genus Orobanche, and in particular in the case proposed by authors on weed Xanthium italicum. The article is well structured and written but deserve of some correction that I have highlighted. I appreciate the original idea of the work which with a few revisions will convince me and the editor to publish it on Journal.

Title

Please consider in the title the scientific name in the complete way with the author name, in this case L. = Linneus. …..Orobanche L. on host weed Xanthium spinosum L.

 done

Introduction

  • Lines 41-42. Please complete the concept with this crucial point and choose 3/4 international references: “The Orobanchaceae family as the largest family of parasitic flowering plants includes nearly 2060 holo- or hemiparasitic species in different genera [1], and are very important also for their conservation because are very rare and are not included in the Crop Wild Relatives lists in many countries [choose 3/4 references, one of these Perrino and Wagensommer 2022].
  • It was corrected
  • Line 42. See my previous comment. In general in the international audience the author of the species is reported only the first time in the paper. So, remember this my annotation for whole manuscript. Orobanche
  • It was corrected

  • Lines 44-45. When the author declare “In Iran there are 36 different Orobanche species [3]”, want to refer only to wild species or togheter cultivated and wild? Please clarify because in some countries Orobanche are also cultivated!

It was corrected

  • Lines 45-48. Please add references for these statements “All plants in the genus Orobanche are root parasites that naturally attach to their host with haustoria formation to extract nutrients (choose reference).

A reference was added

Orobanche species were reported from 58 countries (choose 3/4 references).

One reference was added

  •  Despite widespread distribution of Orobanche members worldwide, yield loss of host crops occurs mainly in the north temperate regions including East and South Europe, West Asia and North Africa (choose reference);

reference were added

  • Line 52. See my previous comments:
  • Orobanche aegyptiaca
  • Line 59. lavandulacea in italic
  • done

  • Line 59. See my previous comment. Bituminaria bituminosa (L.) C.H. Stirt.

 It was corrected

Results

Well done, the figure and tables are clear. Few observations

  • Table 1. Please must be separate genus to species in this way: i.e. Orobanche gratiosa, Orobanche aegyptiaca;
  • done
  •  
  • Line 160. Amova or Anova? Please verify
  •  Amova (Analysis of molecular variance) is correct
  1. Discussion

Few observations

  • Line 205. Space between gymnosperms[22, 23]. done
  •  
  • Line 235. See my previous comments for botanical point of view. Kalanchoe Adanson;

 done

  1. Materials and Methods
  • Please give a geographic coordinates about material collection, of Karaj (Alborz Province, Iran) and if possible a map.

Coordinates and map were added to manuscript.

  •  
  • Line 245. All in italic: …Plant material and sampling; done
  • Lines 246-247. Please check the font and size of characthers of this period: “Plant material of Orobanchaceae from one locality in Karaj, Alborz Province, Iran was collected on a summer annual weed plant Xanthium spinosum”.

done

  • Line 269. Orobanche anatolica

  It was corrected

  1. Conclusions

Please spend two words about future research perspectives on these topic.

Concloution have been amended

  • Line 279. aegeptiaca in italic
  • It was corrected

Reference to be added:

  •   Perrino, E.V.; Wagensommer, R.P. Crop Wild Relatives (CWRs) Threatened and Endemic to Italy: Urgent Actions for Protection and Use. Biology 202211, 193. DOI: 10.3390/biology11020193.
  • done

Reviewer 3 Report

This is an interesting and useful taxonomic study on Orobanche. In this article, the authors described the new Orobanche taxon and performed a detailed phylogenetic analysis of a new isolate of the holoparasitic plant using the sequences of the 5.8S rRNA region. Certainly, these are important results that will be helpful to future investigators. This article is interesting, it focuses on poorly researched issues. The authors are advised to address the following comments for improving the quality of the article.

Suggestions:

Line 19: “rRNAregion” - separate with a space

Line 44: “species(including…)” - separate with a space

Line 46: reference needed

Line 47 and 49: reference needed

Line 80: brackets should not be italicized

line 115: Table1: (1) the abbreviations sp. and cf. do not write in italics; please separate the species names, it should be: Orobanche gratiosa,  Orobanche aegyptiaca, Orobanche krylowii, Orobanche lycoctoni, etc.

Figure 3.  and 5. Conclusions: Species names should be italicized

Line 277-280: I think that a subsection "conclusions" should be supplemented. Please summarize the research results and provide clear taxonomic conclusions. Please explain in the text why the authors used such a strange nomenclature as  Orobanche aegyptiaca ecotype alborzica. What have your results changed in the current classification system?

Author Response

REVIEWER#3

This is an interesting and useful taxonomic study on Orobanche. In this article, the authors described the new Orobanche taxon and performed a detailed phylogenetic analysis of a new isolate of the holoparasitic plant using the sequences of the 5.8S rRNA region. Certainly, these are important results that will be helpful to future investigators. This article is interesting, it focuses on poorly researched issues. The authors are advised to address the following comments for improving the quality of the article.

Suggestions:

Line 19: “rRNAregion” - separate with a space

 It was corrected

Line 44: “species(including…)” - separate with a space

 It was corrected

Line 46: reference needed

 It was corrected

Line 47 and 49: reference needed

 It was corrected

Line 80: brackets should not be italicized

 It was corrected

line 115: Table1: (1) the abbreviations sp. and cf. do not write in italics; please separate the species names, it should be: Orobanche gratiosa,  Orobanche aegyptiaca, Orobanche krylowii, Orobanche lycoctoni, etc.

 It was corrected

Figure 3.  and 5. Conclusions: Species names should be italicized

Line 277-280: I think that a subsection "conclusions" should be supplemented. Please summarize the research results and provide clear taxonomic conclusions. Please explain in the text why the authors used such a strange nomenclature as  Orobanche aegyptiaca ecotype alborzica. What have your results changed in the current classification system?

Conclusions has been changed

Round 2

Reviewer 1 Report

I think that afetr the revision the text is significatly improved and is almost ready for publication. Only a last check by an English native speaker is probably needed

Reviewer 2 Report

Dear Authors, I appreciate the work done in this last version of the manuscript. 

For my opinion the work is now able to be published

Congratulation,

reviewer

Reviewer 3 Report

The authors have adequately addressed my comments and suggestions towards improvement and/or correction. I have no further comments.

I confirmed changes related with my previous comments. I think that this ms is now acceptable.